# SCL-WC: Cross-Slide Contrastive Learning for Weakly-Supervised Whole-Slide Image Classification

**Xiyue Wang**[1,2], **Jinxi Xiang**[3], **Jun Zhang**[3*], **Sen Yang**[3], **Zhongyi Yang**[3], **Minghui Wang**[1,2], **Jing Zhang**[1*], **Wei Yang**[3], **Junzhou Huang**[3], and **Xiao Han**[3]

[1]College of Biomedical Engineering, Sichuan University, Chengdu, China 610065
[2]College of Computer Science, Sichuan University, Chengdu, China 610065
[3]Tencent AI Lab, Shenzhen, China 518057
wangxiyue@stu.scu.edu.cn, wangminghui@scu.edu.cn, jing_zhang@scu.edu.cn,
{jinxixiang, junejzhang, scusenyang, zhongyiyang, willyang, joehhuang, haroldhan}@tencent.com

## Abstract

Weakly-supervised whole-slide image (WSI) classification (WSWC) is a challenging task where a large number of unlabeled patches (instances) exist within each WSI (bag) while only a slide label is given. Despite recent progress for the multiple instance learning (MIL)-based WSI analysis, the major limitation is that it usually focuses on the easy-to-distinguish diagnosis-positive regions while ignoring positives that occupy a small ratio in the entire WSI. To obtain more discriminative features, we propose a novel weakly-supervised classification method based on cross-slide contrastive learning (called SCL-WC), which depends on task-agnostic self-supervised feature pre-extraction and task-specific weakly-supervised feature refinement and aggregation for WSI-level prediction. To enable both intra-WSI and inter-WSI information interaction, we propose a positive-negative-aware module (PNM) and a weakly-supervised cross-slide contrastive learning (WSCL) module, respectively. The WSCL aims to pull WSIs with the same disease types closer and push different WSIs away. The PNM aims to facilitate the separation of tumor-like patches and normal ones within each WSI. Extensive experiments demonstrate state-of-the-art performance of our method in three different classification tasks (e.g., over 2% of AUC in Camelyon16, 5% of F1 score in BRACS, and 3% of AUC in DiagSet). Our method also shows superior flexibility and scalability in weakly-supervised localization and semi-supervised classification experiments (e.g., first place in the BRIGHT challenge). Our code will be available at https://github.com/Xiyue-Wang/SCL-WC.

## 1 Introduction

The gold standard for cancer diagnosis is derived by examining pathological slides. With the advance in scanning technology, tissue slides are scanned into whole-slide images (WSIs) for better management and processing, which facilitates the development of computational pathology [1; 2]. Due to the gigapixel size of WSIs and their wide variations (e.g., tumor types and staining protocols) [3; 4], acquiring exhaustive pixel/patch-level annotations is very time-consuming and expensive. As the dataset size increases, such sufficient labels are obviously impractical. In practice, weak annotations at the WSI level are more readily available in clinical reports, which facilitates the emergence of weakly-supervised WSI classification (WSWC) studies.

---

*Corresponding author

Existing WSWC studies are typically formulated based on multiple instance learning (MIL), which defines each WSI as a bag and patches cropped from the WSI as individual instances [5; 6; 7; 8; 9; 10; 11; 12; 13; 14]. It is noted that a positive bag contains at least one positive instance while a negative bag contains all negatives [15]. The training process in the MIL paradigm encompasses two steps: (i) feature encoding for patches cropped from a WSI and (ii) feature aggregation under the same WSI. For the feature encoding, the majority of recent methods directly adopt ImageNet-pretrained backbone as an off-the-shelf feature extractor [8; 9; 10; 11; 12; 14] and a few studies adopt self-supervised histopathology-pretrained features [13]. For the feature aggregation, deep attention pooling [10; 11; 12; 13], graph neural network [14], and sequence models [5; 8] are used for effective feature aggregation. The deep attention method drives the importance of each patch for the final WSI prediction, generating interpretable results. The graph neural network and sequence models fully consider the intra-WSI context and long-range dependencies.

However, these methods still have two limitations. First, previous feature encoders are either trained on out-of-domain images in a supervised manner or pretrained on limited in-domain data in a self-supervised manner, which is not infeasible to extend to large datasets of histopathological images due to the difficulty in capturing sufficient variability across organs and diseases. Thus, there is a lack of a universal feature extractor trained on large and diverse histopathological images. Second, previous feature aggregators unfortunately fail to explore the inter-WSI separability and ignore the global feature comparisons across the training WSIs, resulting in limited generalizability for WSIs with a small proportion of disease-positive regions.

To enhance the feature discriminative ability for each patch, we propose a novel WSWC method called SCL-WC that aims to achieve both intra-WSI local patch separation and inter-WSI global feature contrast. Specifically, we first apply the MoCo v3 framework [16] to pretrain a Swin Transformer [17] that is then adopted as an offline feature encoder for all patches, which provides a proper initialization to alleviate the over-fitting problem. Then, we design a novel aggregation algorithm that contains three modules, namely, class-specific deep attention (CDA), positive-negative-aware modeling (PNM), and weakly-supervised cross-slide contrastive learning (WSCL). The CDA follows the previous deep attention paradigm to assign a learnable weight for each patch to indicate its contribution to the WSI prediction. The PNM explicitly models the appearance of positive and negative patches within WSIs to capture discriminative feature representations, promoting normal/abnormal tissue separation. The WSCL constructs diverse feature comparisons across WSIs to refine task-specific features, where the WSI-level supervision enables more reliable separation capabilities for each class in the contrastive learning setting, helping capture informative features.

Our contributions can be summarized as follows. (i) Pioneeringly, a novel WSCL module is proposed for global feature contrast across WSIs, which helps extract more distinguishable features to facilitate both inter-class separability and intra-class compactness. (ii) The PNM is designed to explicitly split each WSI feature space into positive and negative subspaces, thus helping exclude uninformative patches. (iii) Our proposed SCL-WC achieves a significant performance gain compared with other WSWC methods. By feasibly extending it to a semi-supervised classification task, our method won first place in the BRIGHT challenge.

## 2 Related work

### 2.1 Weakly-supervised WSI classification

The WSWC task aims to select representative patches to trigger the corresponding WSI-level labels. Currently, MIL has been applied to formulate this problem with remarkable success [5; 6; 7; 8; 9; 10; 11; 12; 13; 14], which requires two key techniques: patch-level feature encoding and feature aggregation for WSI representation.

These feature encoders can be divided into online and offline models, where the online networks require real-time updates [5; 6; 7], resulting in more training epochs to converge compared to the well-pretrained offline models. The utilized offline feature extractors include supervised ImageNet-pretrained [8; 9; 10; 11; 12; 14] and unsupervised histopathological-image-pretrained models [13]. However, the natural images (out-of-domain data) are difficult to accurately capture the textural and morphological characteristics of histopathological images without any fine-tuning. The used self-supervised model is pretrained on a small number of unlabeled samples, resulting in limited feature representations.

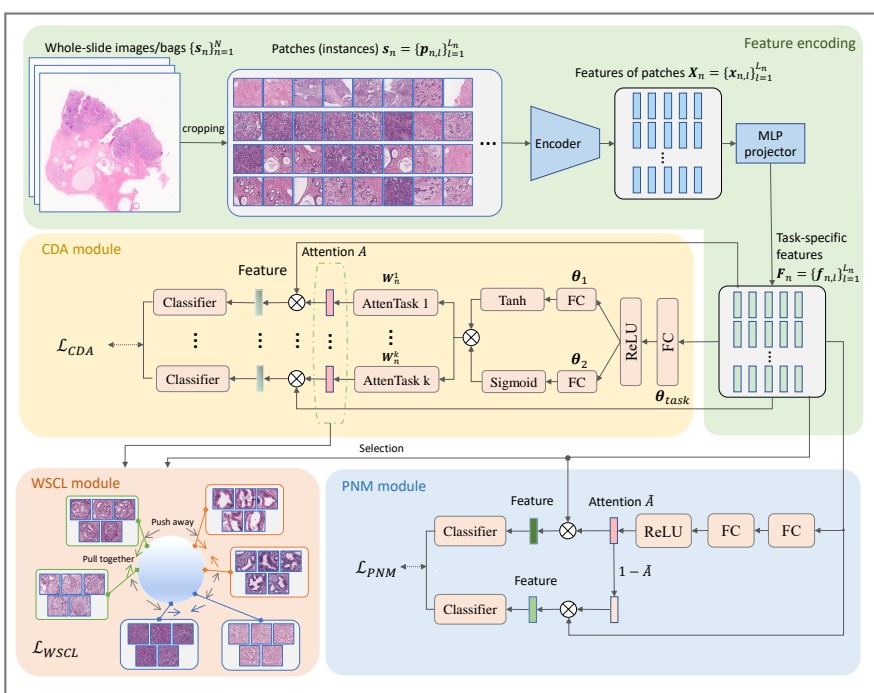

Figure 1: The pipeline of our proposed SCL-WC method, which consists of two parts: feature encoding and task-specific feature aggregation. In the feature encoding process, each WSI (bag) is first cropped into patches that are then encoded by SSL-pretrained encoder (Swin Transformer). We use an MLP projector to map the offline features into task-specific features that will be tuned later. The feature aggregator contains three modules: CDA, PNM, and WSCL.

The utilized feature aggregation algorithms can be divided into two lines: partial-instance-contributed and full-instance-contributed methods. The partial-instance-contributed methods keep a fixed number of patches in each WSI by randomly sampling from tissue regions [7], selecting the top-k patches with high confidence scores [5], or selecting a subset from each patch-level cluster [6; 9]. A small portion of patches may not fully capture the morphological features for each WSI, leading to misdiagnosis and missed diagnoses. The full-instance-contributed methods adopt deep attention pooling [10; 11; 12; 13], graph neural network [14], and sequence model [5; 8] to integrate all patches within a WSI, where patches can be assigned appropriate weight parameters by the network training to drive a WSI-level representation. These methods take into account the contributions of all patches and are more flexible and adaptable to other tasks than previous limited representative patches.

## 2.2 Contrastive representation learning

Contrastive representation learning aims to learn a universal feature by pulling samples belonging to the same class together and pushing samples belonging to different classes apart. The most popular unsupervised contrastive learning methods are SimCLR [18; 19] and MoCo [20; 21; 16], which take data augmentations from the same image as positives and those from different images as negatives. In later supervised contrastive learning, data annotations help to formulate the correct contrastive pairs, resulting in more representative features [22]. Based on previous studies, this work proposes a WSCL module in a MIL setting, aiming to extract class-specific distinguishable features in specific tasks.

## 3  Methods

### 3.1  Definition for the WSWC task

Suppose we have a series of training WSIs $\{\mathbf{s}_n\}_{n=1}^N$ and their corresponding slide-level labels $\{y_n\}_{n=1}^N$, where $y_n \in \{0, 1, \cdots, C\}$ represents the label of the $n^{th}$ slide, $C$ denotes the number of

classes (e.g., cancer subtypes) and 0 corresponds to the normal (negative) tissues. The $n^{th}$ slide can be represented as $\mathbf{s}_n = \{\mathbf{p}_{n,l}\}_{l=1}^{L_n}$, where $\mathbf{p}_{n,l}$ is the $l^{th}$ patches cropped from the slide by a sliding window, $L_n$ denotes the number of patches within the slide, which may vary across slides of different image sizes. It is noted that only slide-level annotations exist and annotations for internal patches are not explicit. The WSWC task aims to train a model with weak labels to conduct WSI-level prediction. The overview of our proposed SCL-WC method is shown in Figure 1, which mainly includes a feature encoding process and a task-specific feature aggregation.

## 3.2 Feature encoding

Self-supervised learning (SSL) has the ability to train a universal feature encoder under the supervision of data itself, which has been widely used in computer vision and medical image analysis [18; 20; 23; 24]. Benefiting from its remarkable success, this work applies MoCo v3 framework [20; 16] to pretrain a feature encoder (Swin-Transformer [17]) on 15 million unlabeled patches from TCGA [25] and PAIP [26] datasets. Then, the well-pretrained backbone is employed as an offline feature extractor to transform patches into a series of $q$-dimensional vectors. For example, the feature vectors of the $n^{th}$ slide can be represented as $\mathbf{X}_n = \{\mathbf{x}_{n,l}\}_{l=1}^{L_n}$, where $\mathbf{x}_{n,l} \in \mathbb{R}^q$ denotes the vector of the $l^{th}$ patch and $\mathbf{X}_n \in \mathbb{R}^{L_n \times q}$ is obtained by stacking all patches within the WSI. Due to the task-agnostic characteristic of these pretrained features, we map them into a task-specific space using fully connected layers and non-linear activation functions. Thus, the task-specific features in the $n^{th}$ slide can be represented as $\mathbf{F}_n = \{\mathbf{f}_{n,l}\}_{l=1}^{L_n} = \mathrm{ReLU}\left(\mathrm{FC}\left(\mathbf{X}_n, \boldsymbol{\theta}_{\text{task}}\right)\right)$, where $\mathbf{F}_n \in \mathbb{R}^{L_n \times d}$, $\boldsymbol{\theta}_{\text{task}}$ is trainable parameters in the fully connected layers.

## 3.3 Task-specific feature aggregation

Our task-specific feature aggregator consists of three modules: CDA, PNM, and WSCL. The CDA aims to parameterize the contribution of each patch to the final WSI prediction, helping provide interpretable results. The PNM is composed of positive-aware loss and negative-aware loss, which is designed to mitigate the noise caused by the large proportion of normal (negative) subregions in each WSI. The WSCL considers global image information by pulling positive bags closer and pushing negative ones away. The combination of these three modules enables the feature aggregator to explore the intra- and inter-WSI complementary information, helping tune the task-specific feature layers for more discriminative patch-level representations and further improving weakly-supervised classification and localization performance.

**Class-specific deep attention.** The CDA module acts as a main branch to aggregate these patch-level features $\mathbf{F}_n = \{\mathbf{f}_{n,l}\}_{l=1}^{L_n}$ into a slide-level vector $\tilde{\mathbf{F}}_n$ using deep attention-based MIL pooling [15] that assigns a weight for each patch within WSI to specify its relative contribution to the final WSI prediction. This deep attention mechanism is complemented by several fully connected layers. We use $A_n^{i,l}$ to denote the weight score of the $l^{th}$ patch in the $n^{th}$ slide for the $i^{th}$ class, which is calculated as

$$A_n^{i,l} = \frac{\exp\left\{\mathbf{W}_n^i\left(\tanh\left(\boldsymbol{\theta}_1 \mathbf{f}_{n,l}^{\top}\right) \odot \mathrm{sigm}\left(\boldsymbol{\theta}_2 \mathbf{f}_{n,l}^{\top}\right)\right)\right\}}{\sum_{j=1}^{L_n} \exp\left\{\mathbf{W}_n^i\left(\tanh\left(\boldsymbol{\theta}_1 \mathbf{f}_{n,j}^{\top}\right) \odot \mathrm{sigm}\left(\boldsymbol{\theta}_2 \mathbf{f}_{n,j}^{\top}\right)\right)\right\}}, \tag{1}$$

where $\boldsymbol{\theta}_1 \in \mathbb{R}^{M \times d}$, $\boldsymbol{\theta}_2 \in \mathbb{R}^{M \times d}$, $\mathbf{W}_n^i \in \mathbb{R}^{1 \times M}$, and $\odot$ denotes element-wise multiplication. The attention score $A_n^{i,l}$ ranges from 0 to 1 and the final attention matrix for each class in each slide is normalized such that the sum of these weights is 1, *i.e.*, $\sum_{l=1}^{L_n} A_n^{i,l} = 1$. The attention matrix for the $n^{th}$ slide is calculated as $A_n \in \mathbb{R}^{L_n \times C}$. And then, the weighted slide-level feature of the $i^{th}$ class can be represented as $\tilde{\mathbf{F}}_n^i \in \mathbb{R}^d$, which is computed by

$$\tilde{\mathbf{F}}_n^i = \sum_{l=1}^{L_n} A_n^{i,l} \mathbf{f}_{n,l}, \tag{2}$$

Next, these features are fed into the $i^{th}$ classifier to drive the corresponding predicted probability for the class. Then, softmax is applied over each class to normalize the probability distribution. We

use $p_n^i$ to represent the probability that the slide belongs to the $i^{th}$ category and $y_n^i$ to denote the ground-truth label of the $n^{th}$ slide in the one-hot form. They are used to calculate the MIL-based slide classification loss $\mathcal{L}_{mil}$ in the min-batch size of $B$ as follows.

$$\mathcal{L}_{mil} = -\frac{1}{B}\sum_{n=1}^{B}\sum_{i=0}^{C} y_n^i \log\left(p_n^i\right), \tag{3}$$

In addition to the $\mathcal{L}_{mil}$, we also consider adding an auxiliary instance discrimination loss to further enhance the class-specific features. Due to the unavailability of the patch-level annotations, we generate pseudo labels of 1 for these high-attention patches (i.e., top-$k$ attention score) and 0 for these low-attention patches (i.e., bottom-$k$ attention score) by sorting the attention scores in a specified column of $A_n$ corresponding to the real class of the WSI, which contains a total of $2k$ samples for the classification task using a linear classifier with binary cross-entropy loss as follows.

$$\mathcal{L}_{ins} = -\frac{1}{2k}\sum_{j=1}^{2k}(y_j \log\left(p_j\right) - (1 - y_j)\log\left(1 - p_j\right)), \tag{4}$$

where $y_j$ and $p_j$ denote the pseudo label and predicted probability of the $j^{th}$ instance, respectively. The final CDA-based loss $\mathcal{L}_{\text{CDA}}$ is the summed as follows: $\mathcal{L}_{\text{CDA}} = \lambda_1 \mathcal{L}_{\text{mil}} + \lambda_2 \mathcal{L}_{\text{ins}}$.

**Positive-negative-aware modeling.** We propose a PNM module to consider the presence of a large number of normal (negative) tissues within positive WSIs, which should be separated from the abnormal (positive) regions as much as possible. To achieve this, we split the feature space of positive WSI into positive and negative subspaces to enable distinguishable feature learning. Specifically, we first characterize the relevance of the $l^{th}$ patch to the slide-level prediction by calculating its class-agnostic weight score $\tilde{A}_n^l$ through two fully connected layers, i.e., $\tilde{A}_n^l = \text{sigm}(\boldsymbol{\theta}_4(\text{ReLU}(\boldsymbol{\theta}_3 \mathbf{f}_{n,l}^\top)))$. Then we use $1 - \tilde{A}_n^l$ to weight the slide-level feature that focuses on the negative subregions. These new weighted slide features are calculated by

$$\overline{\mathbf{F}}_n^{pos} = \frac{1}{L_n}\sum_{l=1}^{L_n} \tilde{A}_n^l \mathbf{f}_{n,l}, \qquad \overline{\mathbf{F}}_n^{neg} = \frac{1}{L_n}\sum_{l=1}^{L_n}(1 - \tilde{A}_n^l)\mathbf{f}_{n,l}, \tag{5}$$

where $\overline{\mathbf{F}}_n^{pos}$ and $\overline{\mathbf{F}}_n^{neg}$ are the weighted slide-level feature for the prediction of positive and negative samples, respectively. $\tilde{A}_n^l$ is the weight score for the $l^{th}$ patch within the WSI, which ranges from 0 and 1. Then the slide-level prediction probability for positive $p_n^{\text{pos}}$ and negative $p_n^{\text{neg}}$ features can be obtained by feeding $\overline{\mathbf{F}}_n^{pos}$ and $\overline{\mathbf{F}}_n^{neg}$ into a fully connected layer with a softmax function. Then, the PNM-based classification loss can be calculated as

$$\mathcal{L}_{\text{PNM}} = -\frac{1}{B}\sum_{n=1}^{B}(\log p_n^{\text{pos}} - \log p_n^{\text{neg}}), \tag{6}$$

where $\mathcal{L}_{\text{PNM}}$ is a summation of two cross-entropy loss functions: positive-aware classification loss and negative-aware classification loss.

**Weakly-supervised cross-slide contrastive learning.** Previous contrastive learning methods applied on histopathological images adopt patches as positive/negative units [13], which ignores the information interaction across slides and captures only local feature representation. Different from them, we propose a WSCL that aims to generate more discriminative class-specific features by comparing feature representations across slides, i.e., pulling slides belonging to the same class closer and pushing slides belonging to different classes away. Due to the huge heterogeneity within each WSI, direct comparisons between WSIs are susceptible to interference from noise, which is instead harmful to network training. Thus, we construct a new series of bags by selecting the most representative patches within each WSI based on the class-specific attention obtained above. Specifically, we use three types of sub-memory banks to store positive, negative, and hard negative bags, respectively. We take the top-$k$ patches within each positive WSI into the positive bags. Similarly, the top-$k$ and bottom-$k$ patches within each negative WSI are inserted into hard negative and negative bags,

respectively. It is noted that samples belonging to different positive categories should be stored in separate positive bags. Based on the above definition, our WSCL-based loss can be defined by

$$\mathcal{L}_{\text{WSCL}} = \sum_{\mathbf{z}_i \in \mathbf{B}} \frac{-1}{k \times |\mathbf{P}|} \sum_{\mathbf{p}_r \in \mathbf{P}} \sum_{\mathbf{z}_j \in \mathbf{p}_r} \log \frac{\exp\left(\mathbf{z}_i \cdot \mathbf{z}_j / \tau\right)}{\sum_{a \in \mathbf{Q}} \exp\left(\mathbf{z}_i \cdot \mathbf{z}_a / \tau\right)}, \tag{7}$$

where $\mathbf{B}$ is an anchor bag, $\mathbf{z}_i$ ($\mathbf{z}_i \in \mathbf{B}$) is the $i^{th}$ anchor patch from the anchor bag. For an anchor bag, $\mathbf{P}$, $\mathbf{N}$, and $\mathbf{H}$ are used to represent its corresponding sets of positive, negative, and hard negative bags, and $|\mathbf{P}|$, $|\mathbf{N}|$, and $|\mathbf{H}|$ are used to represent their corresponding bag number, respectively. $\mathbf{p}_r$ is used to represent the $r^{th}$ bag in the positive set $\mathbf{P}$. $\mathbf{Q} = \mathbf{P} \cup \mathbf{N} \cup \mathbf{H}$ denotes the total bags used in the WSCL calculation process.

The final loss function is the summation of the CDA-based loss, PNM-based loss, and WSCL-based loss, which is computed as $\mathcal{L}_{\text{total}} = \mathcal{L}_{\text{CDA}} + \beta \mathcal{L}_{\text{PNM}} + \gamma \mathcal{L}_{\text{WSCL}}$, where $\beta$ and $\gamma$ are hyper-parameters used to adjust the contribution of each loss.

## 4 Experiments

In this section, we construct a series of experiments on five public histopathological image datasets. The five datasets are collected from three different organs, including prostate (PANDA [27] and DiagSet [28]), breast (Camelyon16 [29] and BRACS [30]), and colorectum (TCGA-CRC-DX), which are detailed in Table 1.

**Datasets.** Camelyon16 is released for metastatic breast cancer detection, which contains 240 WSIs with metastasis and 159 normal WSIs. These WSIs are split into a training set of 270 WSIs and a test set of 129 WSIs by the provider. Although pixel-level annotations are available in the dataset, they are only used to evaluate our weakly-supervised localization performance.

BRACS is released at the BRIGHT challenge for the classification of breast tumor subtyping. We follow the challenge for a 3-class WSI classification: non-cancerous (Non. with 288 WSIs), pre-cancerous (Pre. with 155 WSIs), and cancerous (Can. with 260 WSIs). The challenge organizer splits the total of 703 WSIs into a training set of 423 WSIs, a test-1 set of 80 WSIs, and a test-2 set of 200 WSIs. It is noted that some well-annotated patches (3566) are also provided for training, which are ignored in our WSWC task and are utilized in the training process of semi-supervised classification when participating in the challenge.

PANDA is the largest publicly available WSI data for 2-class prostate cancer classification, which releases a total of 10,616 WSIs (7724 Can. and 2892 Non.). We split them into training, validation, and test sets with a ratio of 7:1:2.

DiagSet contains three subsets of histopathological images for 2-class prostate cancer classification: DiagSet-A, DiagSet-B, and DiagSet-C. These subsets are adopted as three external test sets to demonstrate the model generalizability to unseen data. DiagSet-A, DiagSet-B, and DiagSet-C contain 430 WSIs (Can: 228 WSIs and Non.: 202 WSIs), 4675 WSIs (Can: 2090 and Non.: 2585 WSIs), and 46 WSIs (Can: 37 WSIs and Non.: 9 WSIs), respectively.

TCGA-CRC-DX (The Cancer Genome Atlas colon and rectal cancer) [31] is used to test the performance of our SCL-WC method for microsatellite instability prediction, which contains a total of 428 WSIs (62 WSIs with microsatellite instability and 366 WSIs with microsatellite stability). The data splitting process is kept consistent with the original reference [31], i.e., 4-fold cross-validation based on the publicly released splitting table.

**Evaluation metrics.** For a fair comparison with previous methods, we adopt accuracy (ACC), area under the curve (AUC), and F1 score as metrics to evaluate our weakly-supervised classification performance. Following the Camelyon16 challenge, free response operating characteristic curves (FROC) is used to assess the tumor localization performance [29]. The experimental setups can be seen in the supplementary material.

Table 1: Datasets (No.: number). These WSIs are cropped into patches with a size of $224 \times 224$ pixels and a resolution of 1.0 microns per pixel.

|              | WSI No. | Patch No. |
|--------------|---------|-----------|
| Camelyon16   | 399     | 920,119   |
| BRACS        | 703     | 1,552,263 |
| PANDA        | 10616   | 1,843,968 |
| DiagSet-A    | 430     | 513,274   |
| DiagSet-B    | 4675    | 4,819,345 |
| DiagSet-C    | 46      | 50,845    |
| TCGA-CRC-DX  | 428     | 1,264,344 |
| Total        | 15676   | 9,699,814 |

Table 2: Weakly-supervised classification results on Camelyon16 dataset

| Methods       | ACC      | AUC      | FROC     |
|---------------|----------|----------|----------|
| Full-sup      | 0.9302   | **0.9762** | 0.6543 |
| Human [29]    | /        | 0.9660   | **0.7325** |
| Mean-pooling  | 0.7984   | 0.7620   | 0.1162   |
| Max-pooling   | 0.8295   | 0.8641   | 0.3313   |
| MIL-RNN [5]   | 0.8062   | 0.8064   | 0.3048   |
| ABMIL [15]    | 0.8450   | 0.8653   | 0.4056   |
| DSMIL [13]    | 0.8992   | 0.9165   | 0.4371 |
| CLAM [10]     | 0.8682   | 0.9121   | 0.4104   |
| TransMIL [8]  | 0.8992 | 0.9337 | /   |
| Ours          | **0.9147** | **0.9566** | **0.5659** |

Table 3: Weakly-supervised classification results on BRACS

|              | ACC      | AUC      | F1       |
|--------------|----------|----------|----------|
| Mean pooling | 0.7333   | 0.7294   | 0.2932   |
| Max pooling  | 0.7458   | 0.7992   | 0.4386   |
| ABMIL [15]   | 0.7291   | 0.8055   | 0.4842   |
| TransMIL [8] | 0.7208   | 0.7863   | 0.5602   |
| CLAM [10]    | 0.7583   | 0.8158   | 0.5611   |
| DSMIL [13]   | 0.7644 | 0.8314 | 0.6349 |
| Ours         | **0.8208** | **0.8650** | **0.6886** |

Table 4: Semi-supervised classification results on BRIGHT challenge in terms of F1 score metric

| Rank      | Ave.     | Non.     | Pre.     | Can.     |
|-----------|----------|----------|----------|----------|
| 1 (Ours)  | **0.716** | **0.725** | **0.623** | **0.800** |
| 2         | 0.643 | 0.564 | 0.580 | 0.786 |
| 3         | 0.599    | 0.675    | 0.455    | 0.667    |
| 4         | 0.520    | 0.637    | 0.244    | 0.680    |
| 5         | 0.480    | 0.530    | 0.331    | 0.580    |
| 6         | 0.459    | 0.388    | 0.416    | 0.571    |

## 4.1 Results on Camelyon16 dataset

This subsection validates our weakly-supervised classification and localization capacities on the Camelyon16 dataset by comparing it with state-of-the-art related methods. Detailed results are shown in Table 2, where the best result is bolded, and the second best is underlined. It is noted that, except for the CLAM and TransMIL algorithms, previous methods are all reported in [13], which are implemented using their corresponding official codes along with the SSL-pretrained features by [13]. The CLAM and TransMIL algorithms are implemented by us using their released code and our pretrained features. Thus we can directly compare our aggregation algorithm with these methods.

As shown in Table 2, our method outperforms other WSWC algorithms to a large extent. For example, our method outperforms the previous best-performing TransMIL by around 2% in ACC and 2% in AUC. The reason can be analyzed as follows. In the Camelyon16 dataset, the percentage of anomalous regions within each WSI is typically below 10%. Thus, previous methods are particularly susceptible to noise, resulting in possibly missed detection. Our proposed SCL-WC enables distinctive categorical feature extraction by intra-WSI and inter-WSI complementary information converging, helping alleviate the problem of a small percentage of lesions within WSIs.

The CDA module assigns a learnable weight for each patch to represent its importance for the WSI prediction, which is combined with the feature refinement process in the WSCL and PNM modules to promote better lesion localization ability. The detailed diagnosis-positive localization results are shown in Table 2, Figure 2, and supplementary material. As shown in Table 2, our method achieves an FROC of 0.5659, which outperforms over 10% than other methods and shows the potential to be close to fully-supervised performance with an FROC of 0.6543. In Figure 2 and supplementary material, the warmer colored subregions imply a higher probability of abnormal tissues, which visually demonstrates our superior localization performance even for tiny lesions.

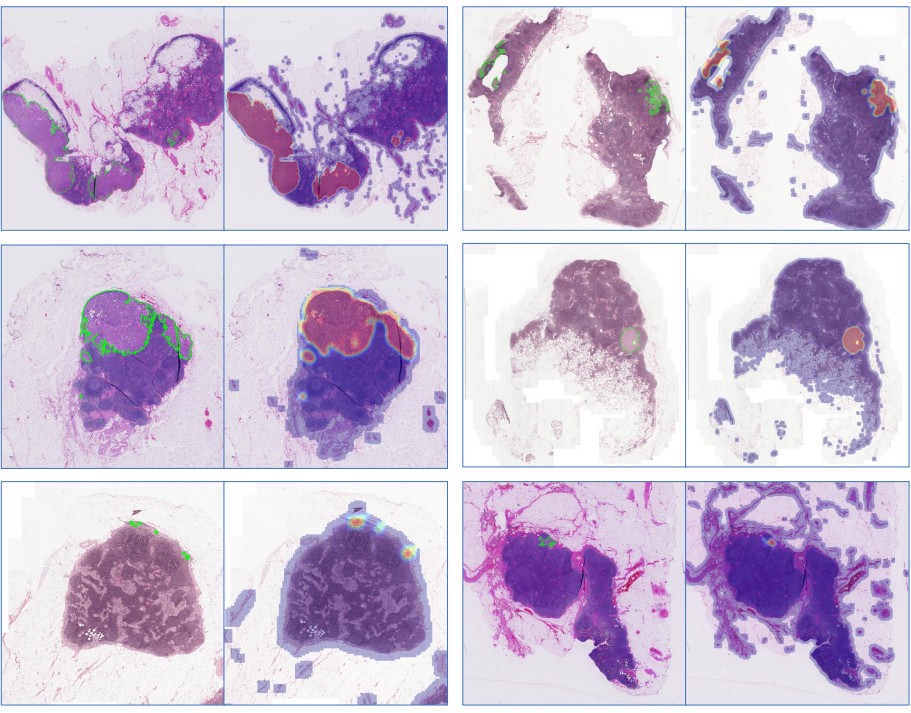

Figure 2: The visualization of weakly-supervised localization results. These six samples are taken from the Camelyon16 dataset. In the image pair, the left image represents the ground truth with green lines to mark lesion regions and the right image is predicted results by our model where these lesion regions are highlighted by warm color.

## 4.2 Results on BRACS, PANDA, and DiagSet datasets

In this subsection, we first conduct 3-class weakly-supervised classification experiments on the BRACS dataset in Table 3. The results of these compared methods are obtained according to their released official codes. The 3-class classification task is very challenging due to the indistinguishable features between some non-cancerous and pre-cancerous regions (e.g., pathological benign V.S. flat epithelial atypia) or between some pre-cancerous and cancerous tissues (e.g., atypical ductal hyperplasia V.S. ductal carcinoma in situ). Thus, the overall classification performance is less than 90% in the current state-of-the-art methods. The previous best-performed method is the DSMIL method in the BRACS dataset, however, it shows suboptimal performance in other datasets (e.g., Table 2 and Table 5). Our proposed SCL-WC remains most effective on all of these datasets, which reflects its stability.

Then, in Table 4, we show our results when participating in the 2022 BRIGHT challenge, which is achieved using a semi-supervised classification scheme extended from our SCL-WC, with little difference in whether a small number of well-annotated patches are used. As shown in Table 4, our method ranked first in the BRIGHT challenge, which shows the high flexibility and scalability of our proposed SCL-WC method.

Next, we validate our algorithm on two larger prostate datasets. Also, extensively external tests are performed to further confirm the robustness and generalization performance of our algorithm. The detailed results are summarized in Table 5, where the PANDA dataset is used for algorithm development and inner validation, and the remaining three subsets of DiagSet are used for external tests. The results of these compared methods are obtained using their released codes. As shown in Table 5, our method consistently performs better on these datasets. Specifically, our method outperforms the CLAM method by around 3% ACC and 2% AUC in the PANDA dataset. In the largest external test set DiagSet-B, our method surpasses DSMIL by around 10% and 3% in ACC and AUC, respectively.

Table 5: Results on the prostate datasets

| | PANDA | | DiagSet-A | | DiagSet-B | | DiagSet-C | |
|---|---|---|---|---|---|---|---|---|
| | ACC | AUC | ACC | AUC | ACC | AUC | ACC | AUC |
| Mean pooling | 0.8407 | 0.9386 | 0.8313 | 0.9237 | 0.8197 | 0.8914 | 0.8478 | 0.9159 |
| Max pooling | 0.8847 | 0.9508 | 0.7330 | 0.9315 | 0.7377 | 0.9371 | 0.8695 | 0.9489 |
| ABMIL [15] | 0.8804 | 0.9514 | 0.7845 | 0.9145 | 0.8032 | 0.9105 | 0.8695 | 0.9609 |
| TransMIL [8] | 0.8715 | 0.9408 | 0.8290 | 0.9290 | 0.8246 | 0.9334 | 0.9130 | 0.9669 |
| DSMIL [13] | 0.8751 | 0.9444 | 0.7072 | 0.9242 | 0.8146 | 0.9431 | 0.8876 | 0.9489 |
| CLAM [10] | 0.8874 | 0.9532 | 0.7822 | 0.9033 | 0.8035 | 0.9051 | 0.8913 | 0.9579 |
| Ours | **0.9194** | **0.9753** | **0.8960** | **0.9560** | **0.9191** | **0.9730** | **0.9565** | **0.9939** |

## 4.3 Comparison between different feature extractors and feature aggregators

The proposed feature extractor is compared with existing feature extractors that are widely used for histopathological image analysis tasks, including an ImageNet-pretrained feature extractor and two other SSL-based feature extractors pretrained on histopathological images (SimCLR and DINO) [32; 33]. To this end, we have conducted new experiments to investigate the performance of our SCL-WC method when using different feature extractors for microsatellite instability prediction on the TCGA-CRC-DX data and breast tumor subtyping on BRACS data. These existing feature extractors have been pretrained by previous studies and can be used as off-the-shelf feature extractors for comparison. The detailed results can be seen in Table 6. It can be seen that the best performance is achieved by combining the proposed feature extractor with the proposed feature aggregator. Comparing different feature extractors tested on TCGA-CRC-DX data, it is also shown that our feature extraction method outperforms the previous best-performing DINO-ViT by around 7% using both aggregation methods.

Table 6 also shows the effectiveness of the proposed feature aggregation scheme by keeping the same feature extractor while testing different feature aggregators (CLAM and ours). Comparing the corresponding columns of TCGA-CRC-DX data in Table 6, it is seen that our aggregation scheme outperforms that of the CLAM method by around 4% for any of the four different feature extractors.

Table 6: Results of weakly supervised classification using different feature extractors and feature aggregators. (Datasets: TCGA-CRC-DX and BRACS; Metric: AUC)

| Feature extractors | Feature aggregators | | | |
|---|---|---|---|---|
| | CLAM | Ours | CLAM | Ours |
| | TCGA-CRC-DX | | BRAC | |
| ImageNet-ResNet50 | 0.7568 | 0.7806 | 0.8158 | 0.8446 |
| SimCLR-ResNet18 [32] | 0.7716 | 0.8108 | 0.7634 | 0.8008 |
| DINO-ViT [33] | 0.7853 | 0.8246 | 0.8108 | 0.8345 |
| Ours-SwinTransformer | 0.8530 | 0.8954 | 0.8378 | 0.8650 |

## 4.4 Ablation study

We construct a set of ablation experiments in Table 7 to demonstrate the effectiveness of key components in our proposed SCL-WC method, including SSL-based feature extractor, $\mathcal{L}_{\mathrm{CDA}}$, $\mathcal{L}_{\mathrm{PNM}}$, and $\mathcal{L}_{\mathrm{WSCL}}$ (with and without hard negatives). As shown in Table 7, the results in the first two rows show that, compared to the ImageNet-pretrained network, the SSL-based histopathology-pretrained feature extractor improves around 3% on Camelyon16 and 2% on BRACS in terms of AUC. As shown in the second and third rows of Table 7, our positive-negative-aware loss brings a consistent 1% performance gain across all datasets and metrics, which verifies the importance of the PNM module. The effectiveness of the WSCL module can be seen in the last three rows of Table 7. When hard-negative samples are not considered, the performance is improved by around 2% in both datasets compared to the results on the third row, and the performance improves further when hard negative samples are added as shown in the last row of Table 7.

Table 7: Results of ablation study

| | Camelyon16 | | BRACS | |
|---|---|---|---|---|
| | ACC | AUC | ACC | AUC |
| ImageNet + $\mathcal{L}_{\mathrm{CDA}}$ | 0.8370 | 0.8730 | 0.7523 | 0.8114 |
| SSL + $\mathcal{L}_{\mathrm{CDA}}$ | 0.8759 | 0.9080 | 0.7635 | 0.8335 |
| SSL + $\mathcal{L}_{\mathrm{CDA}}$ + $\mathcal{L}_{\mathrm{PNM}}$ | 0.8814 | 0.9190 | 0.7725 | 0.8403 |
| SSL + $\mathcal{L}_{\mathrm{CDA}}$ + $\mathcal{L}_{\mathrm{PNM}}$ + $\mathcal{L}_{\mathrm{WSCL}}$ (P+N) | 0.9069 | 0.9330 | 0.7926 | 0.8512 |
| SSL + $\mathcal{L}_{\mathrm{CDA}}$ + $\mathcal{L}_{\mathrm{PNM}}$ + $\mathcal{L}_{\mathrm{WSCL}}$ (P+N+HN) | **0.9147** | **0.9566** | **0.8208** | **0.8650** |

## 5 Conclusion

We propose a novel WSWC method called SCL-WC, which is constructed based on a domain-specific SSL feature extractor and a task-specific feature aggregator. The feature aggregator includes three effective modules: CDA, PNM, and WSCL, which are combined for discriminative patch-level feature refinement, providing not only interpretable results but also fine lesion localization. The proposed SCL-WC method outperforms state-of-the-art WSWC studies over five publicly available datasets for the binary/multiple classification tasks, which also shows good feasibility and scalability in the weakly-supervised localization and semi-supervised classification tasks. However, our method should be extensively validated in larger cohorts from real-world clinical settings before its deployment, and we will explore this in future work.

## 6 Acknowledgment

This research was in part funded by the Science & technology department of Sichuan Province (No. 2020YFG0081) and the National Key R&D Program of China (No. SQ2022YFC2400174).

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
