# SCL-WC: Cross-Slide Contrastive Learning for Weakly-Supervised Whole-Slide Image Classification (Supplementary Materials)

**Xiyue Wang**[1,2], **Jinxi Xiang**[3], **Jun Zhang**[3*] **Sen Yang**[3], **Zhongyi Yang**[3], **Minghui Wang**[1,2], **Jing Zhang**[1*] , **Wei Yang**[3], **Junzhou Huang**[3], and **Xiao Han**[3]

[1]College of Biomedical Engineering, Sichuan University, Chengdu, China 610065
[2]College of Computer Science, Sichuan University, Chengdu, China 610065
[3]Tencent AI Lab, Shenzhen, China 518057

## 1   Overview

The supplementary materials contain eight aspects:

- Links to the datasets used in this work.

- Details during SSL pretraining.

- Difference between our attention mechanism and state-of-the-art methods.

- Detailed experimental setups of our SCL-WC method.

- More visualization results for weakly-supervised localization.

- More detailed results with standard deviation on these prostate datasets.

- Weakly-supervised classification results on two subsets of TCGA.

- A broader impact statement.

## 2   Data link

This work utilizes six datasets, where TCGA and PAIP are used for self-supervised pretraining, and the remaining datasets (Camelyon16, BRACS, PANDA, DiagSet, TCGA-CRC-DX, and TCGA-NSCLC) are used for the weakly-supervised classification procedure. Links to these datasets are provided in Table 1.

Table 1: Links for datasets

| Datasets | Link |
| --- | --- |
| Camelyon16 | https://camelyon16.grand-challenge.org/ |
| BRACS | https://research.ibm.com/haifa/Workshops/BRIGHT/ |
| PANDA | https://panda.grand-challenge.org/ |
| DiagSet | https://ai-econsilio.diag.pl |
| TCGA | https://portal.gdc.cancer.gov/projects |
| PAIP | http://www.wisepaip.org/paip |

---

*Corresponding author

36th Conference on Neural Information Processing Systems (NeurIPS 2022).

# 3 Details during SSL pretraining

## 3.1 Intuition behind the self-supervised feature extractor

We choose the Swin Transformer as the backbone model due to the following considerations: (1) the Swin Transformer is considered as a generally applicable and the state-of-the-art backbone model for many computer vision problems. It contains a hierarchical ViT structure with local self-attention at each layer, which enables it to effectively extract multi-scale features both locally and from a global context [1]. It has been widely applied and shown to outperform CNN models in a variety of applications in both computer vision and medical imaging fields, such as image classification, semantic segmentation, and object detection [1; 2]. (2) For histopathological image feature learning, it is also important to capture both local (cell-level structures) and global (tissue-level contexts) information [3]. Thus, the Swin Transformer model is a very suitable choice.

We choose MoCo v3 as the self-supervised framework for two main reasons. First, MoCo v3 has been proven to outperform other self-supervised learning methods, such as the famous CNN-based SSL frameworks (e.g., SimCLR) [4] and the Transformer-based SSL frameworks (e.g., DINO) [5; 6]. Second, it helps improve the stability of Transformer-type model pretraining and guarantees better performance [4]. We also conducted some comparison experiments to verify the advantages of the MoCo v3 method, which can be seen in Table 2

Table 2: The comparison experiments for different self-supervised frameworks. Note that these methods keep the same Swin Transformer backbone model.

| Methods | Camelyon16 | | BRACS | |
|---|---|---|---|---|
| | ACC | AUC | ACC | AUC |
| SimCLR [7] | 0.8837 | 0.9344 | 0.7625 | 0.8488 |
| DINO [8] | 0.8992 | 0.9464 | 0.7875 | 0.8523 |
| MoCo v3 (Ours) | **0.9147** | **0.9566** | **0.8208** | **0.8650** |

## 3.2 Implementation details of MoCo v3 framework

MoCo v3 uses two parallel branches (online and target branches) to learn meaningful feature representations by solving a pretext task of instance discrimination. These two branches have similar encoder models ($f_q$ and $f_k$) and MLP projectors ($h_q$ and $h_k$). The $f_k$ is momentum updated from $f_q$. The target branch has another MLP projector $g_k$ following $h_k$. Given an image, its two augmented views are used as inputs to the two branches, which are then encoded and projected by the corresponding feature encoders and MLP projectors to obtain the final feature vectors $z_q$ and $z_k$. A contrastive loss is used to pull features obtained from the same image (e.g., $z_q$ and $z_k$) together and push apart features obtained from different images. In our work, we use the Swin Transformer as the backbone model. The first MLP projector ($h_q$ and $h_k$) and the second MLP projector ($g_k$) have three and two linear layers, respectively.

During pretraining, all WSIs in TCGA and PAIP are used as the training data, including frozen and diagnostic (formalin-fixed paraffin-embedded) slides. TCGA contains over 30,000 WSIs covering over 25 anatomical sites and 32 cancer subtypes[2]. PAIP contains 2457 WSIs covering six cancer types [9]. These WSIs are cropped into patches with a size of $512 \times 512$ pixels at $10\times$ magnification, generating approximately 15 million patches. Our data augmentation strategies consider the characteristics of the histopathological images, including random cropping, Gaussian blur, and hue and saturation shifting in the HSV color space [10].

# 4 Differences of attention pooling methods

Among state-of-the-art weakly supervised WSI classification methods, attention pooling is an aggregation scheme widely used in the methods of ABMIL [11], DSMIL [12], and CLAM [13]. The ABMIL method is the first proposer of the attention pooling aggregator, which learns a class-agnostic

---

[2]http://cancergenome.nih.gov/

weight for each patch and then performs a weighted element-wise summation to derive the final features. DSMIL is an extended version of the ABMIL method, which is improved by using different patch weights. It first uses a linear classifier to find a key patch with the highest score and then uses a distance measure between every patch and the key patch to obtain the weight for each patch. CLAM extends the ABMIL method to general multi-class weakly supervised classification tasks using a parallel attention structure for each class. It also adds an SVM-based instance classification branch to improve the instance discrimination ability. Even though these methods achieve superior performance, they never consider the global feature comparison across the training WSIs and the separation between positive and negative patches that are largely unbalanced within positive WSIs.

In our method, although the class-specific deep attention (CDA) module is similar to the CLAM method but with a different instance classification loss, our main contribution is the introduction of the positive-negative-aware modeling (PNM) and weakly-supervised cross-slide contrastive learning (WSCL) modules. The two novel modules fully consider the unique characteristics of WSIs to address the two limitations mentioned above. PNM uses the class-agnostic weight scores to separate the normal and abnormal (positive) regions within positive WSIs as much as possible. WSCL has no attention mechanism, but it helps to explore the complementary information within and across WSIs to obtain more discriminative feature representations, further improving the accuracy of the attention scores.

## 5    Experimental setups

All WSIs are first preprocessed with the Otsu method [14] to remove blank non-tissue regions. In our self-supervised pretraining, the batch size is set to 2048 and each patch is encoded as a vector of length 768. Adam [15] optimizer with an initial learning rate of 1e-4 is adopted. The learning rate is then scheduled by a cosine annealing scheme [16]. Our self-supervised pretraining is performed using 48 Nvidia V100 GPUs. In our weakly-supervised classification procedure, each WSI is cropped into nonoverlapping patches with a size of $224 \times 224$ pixels and a resolution of 1.0 microns per pixel (at around $10\times$). The batch size is set to 1 bag. Adam optimizer is employed with an initial learning rate of 1e-3 (with cosine annealing) and a weight decay of 1e-5. Following the previous contrastive learning method [4], the parameter $\tau$ is set to 0.07 in the WSCL. Following [13], the $k$ in Eq. (4) and (7) is set to 8, and the two weight parameters ($\lambda_1$ and $\lambda_2$) in the $\mathcal{L}_{\mathrm{CDA}}$ are set to 0.8 and 0.2, respectively. The $\beta$, and $\gamma$ in the $\mathcal{L}_{\mathrm{total}}$ are set to 0.2, and 0.01 according to an ablation experiment, which is shown as follows.

Table 3: Effect of hyperparameters $\beta$ and $\gamma$ on classification accuracy of the Camelyon16 dataset (AUC)

| $\beta$ | $\gamma$ | | |
|---|---|---|---|
| | 0.001 | 0.01 | 0.1 |
| 0.02 | 0.9372 | 0.9464 | 0.9418 |
| 0.2 | 0.9497 | **0.9566** | 0.9464 |

## 6    Weakly-supervised localization results

We also provide more examples to demonstrate the localization performance of our weakly-supervised classification algorithm, which is shown in Figure 1. As shown in Figure 1, our SCL-WC method achieves excellent localization results even for small regional lesions.

## 7    Weakly-supervised classification results on the prostate datasets

We further provide detailed weakly-supervised classification results including means and standard deviations for two prostate datasets (PANDA and DiagSet), which are shown in Table 4.

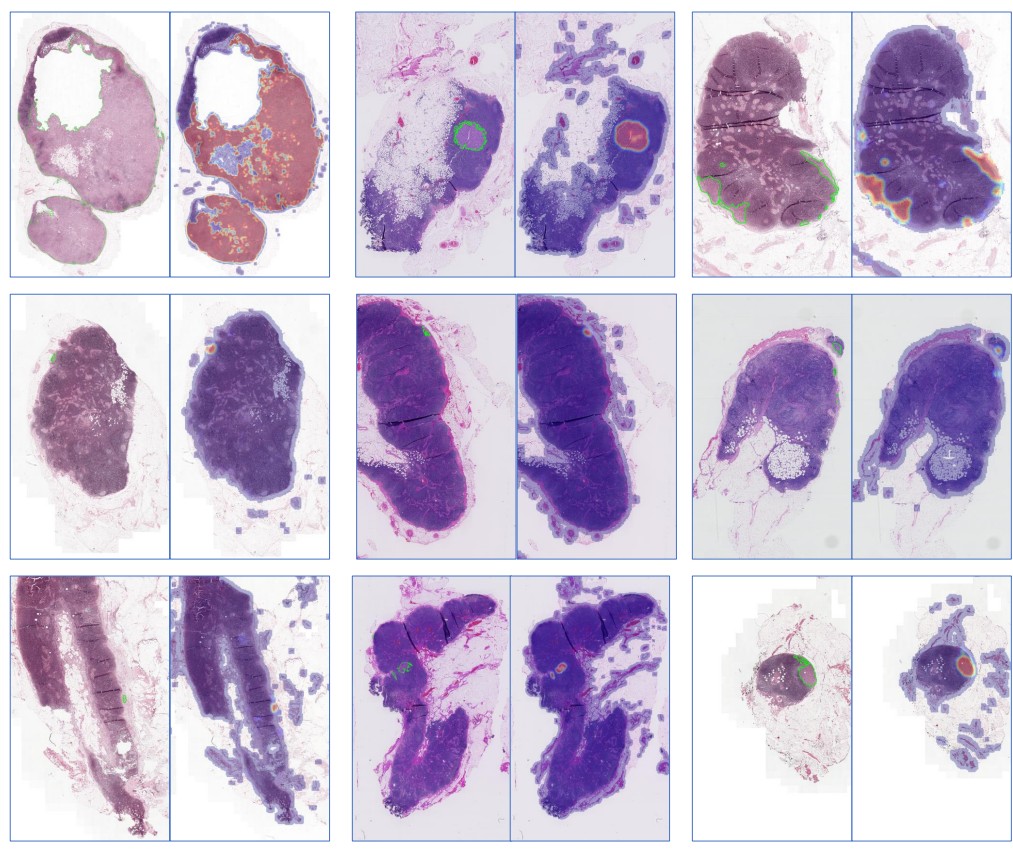

Figure 1: Visualization of weakly-supervised localization results. These samples are taken from the Camelyon16 dataset. In each image pair, the image on the left represents the ground truth of lesion regions marked with green lines and the image on the right is the results predicted by our model where these lesion regions are highlighted in warm colors.

Table 4: Results on the prostate datasets

|  | PANDA | | DiagSet-A | | DiagSet-B | | DiagSet-C | |
|---|---|---|---|---|---|---|---|---|
|  | ACC | AUC | ACC | AUC | ACC | AUC | ACC | AUC |
| Mean pooling | 0.8407±0.0158 | 0.9386±0.0115 | 0.8313±0.0133 | 0.9237±0.0046 | 0.8197±0.0112 | 0.8914±0.0062 | 0.8478±0.0379 | 0.9159±0.0099 |
| Max pooling | 0.8847±0.0026 | 0.9508±0.0047 | 0.7330±0.0758 | 0.9315±0.0187 | 0.7377±0.0809 | 0.9371±0.0204 | 0.8695±0.0307 | 0.9489±0.0306 |
| ABMIL [11] | 0.8804±0.0127 | 0.9514±0.0048 | 0.7845±0.0346 | 0.9145±0.0192 | 0.8032±0.0426 | 0.9105±0.0251 | 0.8695±0.0106 | 0.9609±0.0208 |
| TransMIL [17] | 0.8715±0.0052 | 0.9408±0.0035 | 0.8290±0.0360 | 0.9290±0.0245 | 0.8246±0.0428 | 0.9334±0.0349 | 0.9130±0.0253 | 0.9669±0.0147 |
| DSMIL [12] | 0.8751±0.0122 | 0.9444±0.0046 | 0.7072±0.0691 | 0.9242±0.0175 | 0.8146±0.0824 | 0.9431±0.0161 | 0.8876±0.0162 | 0.9489±0.0182 |
| CLAM [13] | 0.8874±0.0060 | 0.9532±0.0050 | 0.7822±0.0122 | 0.9033±0.0129 | 0.8035±0.0133 | 0.9051±0.0146 | 0.8913±0.0173 | 0.9579±0.0166 |
| Ours | **0.9194±0.0034** | **0.9753±0.0038** | **0.8960±0.0109** | **0.9560±0.0159** | **0.9191±0.0236** | **0.9730±0.0123** | **0.9565±0.0134** | **0.9939±0.0034** |

## 8    Weakly supervised classification results on two subsets of TCGA

To further validate the generalization of our method, we conduct experiments on two subsets of TCGA (i.e., TCGA-CRC-DX for microsatellite instability prediction of colorectal cancer and TCGA-NSCLC for lung tumor subtyping).

TCGA-NSCLC is a subset of the TCGA dataset, which has been used for lung cancer subtyping: lung squamous cell carcinoma (TCGA-LUSC) and lung adenocarcinoma (TCGA-LUAD). It contains a total of 1043 diagnostic WSIs, including 531 WSIs with LUAD and 512 WSIs with LUSC. These WSIs are randomly divided into training, validation, and test sets at the patient level in the ratio of 65:10:25 [17].

The detailed results are shown in Table 5. Results of the compared methods on TCGA-CRC-DX are obtained using their officially released implementations, while the results of the methods on

TCGA-NSCLC are directly copied from their respective publications. It is seen that our method offers state-of-the-art performance on these two new tasks as well. The results indicate that our method is generally applicable and has the potential to be extended to other weakly-supervised applications, such as survival or treatment outcome predictions in computational pathology.

Table 5: Results of weakly-supervised classification on TCGA-CRC-DX dataset for microsatellite instability prediction and TCGA-NSCLC dataset for lung tumor subtyping (AUC).

| Methods | TCGA-CRC-DX | TCGA-NSCLC |
| --- | --- | --- |
| ABMIL [11] | 0.7508 | 0.9200 |
| CLAM [13] | 0.7568 | 0.9380 |
| DSMIL [12] | 0.7630 | 0.9580 |
| TransMIL [17] | 0.7668 | 0.9600 |
| Ours | **0.8954** | **0.9710** |

## 9 Broader impact

Our histopathology-specific feature extractor is pretrained in a self-supervised manner on 15 million unlabeled patches, which guarantees data diversity and can be employed as an offline encoder for various histopathological image applications, promoting the development of computational pathology. Also, our proposed SCL-WC algorithm shows excellent flexibility and scalability, with great potential to be extended to weakly-supervised localization and semi-supervised classification tasks. However, our method should never be used for diagnosis alone, but only to assist pathologists in making faster and better decisions by directly pointing out possible lesion regions. Future work will be dedicated to conducting more clinical verification.