# OpenReview forum: "SCL-WC: Cross-Slide Contrastive Learning for Weakly-Supervised Whole-Slide Image Classification"
_NeurIPS.cc/2022/Conference — NeurIPS 2022 Accept_

### Official Review · Reviewer_nwVH · 2022-07-10

**Rating:** 7
**Confidence:** 4
**Soundness:** 3 good
**Presentation:** 4 excellent
**Contribution:** 3 good

**Summary:**

This paper presents a weakly supervised contrastive learning approach for the classification of whole-slide images (WSI). Specifically, the proposed method consists of two main steps: 1) a domain-specific self-supervised feature extractor pretrained using a Swin-Transformer, and 2) task-specific feature aggregation modules to achieve distinguishable features that can facilitate both inter-class separability and intra-class compactness for WSI representation. The proposed method has been validated on four publicly available large-scale datasets, across two different organs, including prostate (PANDA and DiagSet) and breast (Camelyon16 and BRACS), for binary and multi-class classification tasks. The proposed technique shows significant improvement over the other weakly supervised benchmark methods across all four datasets, with several ablation experiments that demonstrate the efficacy of the proposed idea.

Overall, in my opinion, the idea is impressive, and it is novel in the context of weakly-supervised learning for WSI classification.

**Questions:**

I have a few minor concerns with this paper:

— In Section 3.3, Equation 4 (instance-level loss), it is unclear how these pseudo-labels are obtained from the attention network since we don’t have any patch-level labels in a weakly-supervised setting. Further, what is the chosen value of top-K and bottom-k in the experimentation for identifying high-attention and low-attention patches?

— For SSL pretraining, was the method trained on complete TCGA datasets across multiple organs, or was it trained on a specific subset of TCGA. Please, clarify this in the paper.

**Limitations:**

Yes, the authors have adequately addressed the limitations and potential negative societal impact of their work

**Strengths And Weaknesses:**

 — A series of novel feature aggregation modules (such as class-specific deep attention (CDA), positive-negative-aware modeling (PNM), and weakly-supervised cross-slide contrastive learning (WSCL)) presented in this paper is a clever and well-engineered technique that captures both intra- and inter-WSI feature separability. This strategy is very impressive and generic, which can be extended well beyond other weakly-supervised applications such as survival predictions or treatment outcomes in computational pathology.

— Comparison against a robust set of baseline methods (such as CLAM and Trans-MIL) and ablation experiments that have been conducted demonstrates the usefulness of the proposed technique in challenging four weakly-supervised WSI classification tasks.

— Overall, the paper is well written and easy to follow the core idea.

---

> ### Author Response · Authors · 2022-08-02
> **Response to Reviewer nwVH**
>
> We thank the reviewer for the careful review and valuable feedback. We are very pleased to see that the reviewer finds our method novel, impressive, and generic. We apologize for the lack of clarity on some parts of our original manuscript, as pointed out by the reviewer. We have added more details and explanations in the one-by-one responses below and will also revise the manuscript accordingly. We hope these responses can help clarify the issues.
>
> **In Section 3.3, Equation 4 (instance-level loss), it is unclear how these pseudo-labels are obtained from the attention network since we don’t have any patch-level labels in a weakly-supervised setting. Further, what is the chosen value of top-K and bottom-k in the experimentation for identifying high-attention and low-attention patches?**
>
> We apologize for the confusion in our original writing. To clarify these issues, we have added a concise description to show how the pseudo-labels are obtained in Eq. (4), which can be seen in the paragraph before Eq. (4) in the revised manuscript. In addition, the determination for the parameter $k$ has also been explained in Section 5 in the Appendix. These detailed changes are also shown below.
>
> (1) In the CDA module, the attention matrix of the $n^{th}$ slide is calculated as $A_n \in \mathbb{R}^{L_n\times C}$. Due to the unavailability of the patch-level annotations, we generate pseudo labels of 1 for these high-attention patches (i.e., top-$k$ attention score) and 0 for these low-attention patches (i.e., bottom-$k$ attention score) by sorting the attention scores in a specified column of $A_n$ corresponding to the real class of the WSI.
>
> (2) Following [i], the $k$ in Eq. (4) and (7) is set to 8.
>
> **For SSL pretraining, was the method trained on complete TCGA datasets across multiple organs, or was it trained on a specific subset of TCGA. Please, clarify this in the paper.**
>
> We apologize for the confusion in our original writing. We have added more details during the pretraining, which can be seen in Section 3.2 in the Appendix and also below.
>
> During pretraining, all WSIs in TCGA and PAIP are used as the training data, including frozen and diagnostic (formalin-fixed paraffin-embedded) slides. TCGA contains over 30,000 WSIs covering over 25 anatomical sites and 32 cancer subtypes [ii]. PAIP contains 2457 WSIs covering six cancer types [iii]. These WSIs are cropped into patches with a size of $512 \times 512$ pixels at 10$\times$ magnification, generating approximately 15 million patches. Our data augmentation strategies consider the characteristics of the histopathological images, including random cropping, Gaussian blur, and hue and saturation shifting in the HSV color space [iv].
>
> References:
>
> [i] M. Y. Lu, D. F. Williamson, T. Y. Chen, R. J. Chen, M. Barbieri, and F. Mahmood, “Data efficient and weakly supervised computational pathology on whole-slide images,” Nature Biomedical Engineering, vol. 5, no. 6, pp. 555–570, 2021.
>
> [ii] The Cancer Genome Atlas Project (TCGA). Available from: http://cancergenome.nih.gov/.
>
> [iii] Y. J. Kim, H. Jang, K. Lee, S. Park, S. G. Min, C. Hong, J. H. Park, K. Lee, J. Kim, W. Hong et al., “PAIP 2019: Liver cancer segmentation challenge,” Medical Image Analysis, vol. 67, p. 101854, 2021.
>
> [iv] D. Tellez, G. Litjens, P. Bándi, W. Bulten, J. M. Bokhorst, F. Ciompi, and J. Van Der Laak, “Quantifying the effects of data augmentation and stain color normalization in convolutional neural networks for computational pathology,” Medical Image Analysis, vol. 58, p. 101544, 2019.

---

### Official Review · Reviewer_s5WU · 2022-07-11

**Rating:** 3
**Confidence:** 5
**Soundness:** 2 fair
**Presentation:** 1 poor
**Contribution:** 2 fair

**Summary:**

Summary:
This paper explores the multiple instance learning problems, i.e. the whole slide image classification task. The authors proposed a weakly supervised WSI classification method using cross-slide contrastive learning. Their presented methodology is dependent on self-supervised feature pre-extraction and task-specific weakly-supervised feature refinement and aggregation for WSI-level prediction. The main goal of the weakly-supervised cross-slide constructive learning module is to pull WSIs with the same disease types closer and push different WSIs away.



**Questions:**

The authors need to evaluate the performance of their approach on different cancer indications. For example, there are 32 different cancer indications. I would like to see if your model still performs well enough on a large proportion of those indications.

In addition, the implementation specifications are missing in the manuscript.

The methodology part is not well described and I cannot see its comparison with state-of-the-art. I would suggest the authors provide more clarity on what is the main contribution of this work and how it is different from others.

**Limitations:**


The authors have used a pre-trained feature extractor, i.e. swin transformer from unlabeled patches from TCGA and PAIP, however, they have provided neither intuition behind their choice of feature extractor network compared to other approaches such as SimCLR nor provided their class-specific distribution nor patch sizes, etc. It is unclear to the reviewer why the authors have not provided how well the pre-trained feature extractor has worked in comparison to existing pre-trained models. I would suggest authors take a systematic comparison between different components of their model, for example, compare your model, WSCL using different feature extractors, and empirically demonstrate the performance of your choice of feature extractor. Table 6 could have been extended ImageNet (P+N+HN) analysis as well.
In addition, the pre-trained model is on TCGA slides whereas they have evaluated their model performance on Camelyon16, PANDA, BRACS, and DiagSet. While these datasets are relatively small, why there is no evaluation on the TCGA database? The authors could have demonstrated the generalizability of their approach on larger cancer indications from TCGA instead of focusing only on prostate cancer and breast cancer.
What is the difference between the utilized class-specific deep attention (section 3.3) and the presented attention pooling mechanism in reference 15: Attention-based Deep Multiple Instance Learning? Equation 1 in the manuscript is the same as equation 9 in reference 15.
In table 1. what are the patch sizes? what is the magnification level of your readings? There is a very limited description regarding the implementation specifications, which would make the readers more confused.
Authors could have benefited from a better comparison of their presented approach with its counterparts. For example, they could have described how their attention pooling is different from other state-of-the-art methodologies. In addition, the comparisons between their evaluations with other studies are limited. It is unclear how well their approach is generalizable to other cancer indications.

**Strengths And Weaknesses:**

Strengths:
The authors have investigated one of the challenging research questions in the field, i.e. whole-slide image classification.
The authors have demonstrated the superior performance of their approach to breast cancer and prostate cancer.

Weaknesses:
The authors have used a pre-trained feature extractor, i.e. swin transformer from unlabeled patches from TCGA and PAIP, however, they have provided neither intuition behind their choice of feature extractor network compared to other approaches such as SimCLR nor provided their class-specific distribution nor patch sizes, etc. It is unclear to the reviewer why the authors have not provided how well the pre-trained feature extractor has worked in comparison to existing pre-trained models. I would suggest authors take a systematic comparison between different components of their model, for example, compare your model, WSCL using different feature extractors, and empirically demonstrate the performance of your choice of feature extractor. Table 6 could have been extended ImageNet (P+N+HN) analysis as well.
In addition, the pre-trained model is on TCGA slides whereas they have evaluated their model performance on Camelyon16, PANDA, BRACS, and DiagSet. While these datasets are relatively small, why there is no evaluation on the TCGA database? The authors could have demonstrated the generalizability of their approach on larger cancer indications from TCGA instead of focusing only on prostate cancer and breast cancer.
What is the difference between the utilized class-specific deep attention (section 3.3) and the presented attention pooling mechanism in reference 15: Attention-based Deep Multiple Instance Learning? Equation 1 in the manuscript is the same as equation 9 in reference 15.
In table 1. what are the patch sizes? what is the magnification level of your readings? There is a very limited description regarding the implementation specifications, which would make the readers more confused.
Authors could have benefited from a better comparison of their presented approach with its counterparts. For example, they could have described how their attention pooling is different from other state-of-the-art methodologies. In addition, the comparisons between their evaluations with other studies are limited. It is unclear how well their approach is generalizable to other cancer indications.

---

> ### Author Response · Authors · 2022-08-02
> **Response to Reviewer s5WU (1/5)**
>
> We thank the reviewer for carefully reviewing our manuscript and providing many constructive comments. The main criticisms from the reviewer are the lack of reasoning for choosing different components of the method and insufficient comparison and validation experiments. We would like to mention that some experimental results were excluded in our original manuscript due to paper length considerations. We also would like to emphasize that even though the original validation results only contain two cancer types, breast and prostate, the validation datasets are in fact very large, containing 15676 WSI in total. But in response to the reviewer’s comments, we have added more experimental results and validation and comparison studies. Below are our one-by-one responses, which we hope can adequately address the original concerns.
>
> **The authors need to evaluate the performance of their approach on different cancer indications.**
>
> We agree with the reviewer that evaluating on more cancer types will be more convincing. On the other hand, we would like to emphasize that even though we only reported results on two cancer types, their WSI data are actually very large comparing to the literature, containing 15676 WSIs in total. We use these data for fair and direct comparison with the literature since most existing methods were tested on the same data.
>
> To address the concern of the reviewer and due to time restriction, we have added two new weakly supervised WSI classification experiments on two more cancer types (i.e., TCGA-CRC-DX for microsatellite instability prediction of colorectal cancer and TCGA-NSCLC for lung tumor subtyping). These data have been used in previously published weakly-supervised learning studies [1][2]. The comparison results are shown in Table 5 in the Appendix and also below. Results of the compared methods on TCGA-CRC-DX are obtained using their officially released implementations, while the results of the methods on TCGA-NSCLC are directly copied from their respective publications. It is seen that our method offers state-of-the-art performance on these two new tasks as well. The results indicate that our method is generally applicable and has the potential to be extended to other weakly-supervised applications, such as survival or treatment outcome predictions in computational pathology.
>
> Table 5. Results of weakly supervised classification on TCGA-CRC-DX dataset for microsatellite instability prediction and TCGA-NSCLC dataset for lung tumor subtyping (AUC).
> | Methods | TCGA-CRC-DX | TCGA-NSCLC |
> | :--- | :--- | :--- |
> | ABMIL [3] | 0.751 | 0.920 |
> | CLAM [4] | 0.757 | 0.938 |
> | DSMIL [5] | 0.763 | 0.958 |
> | TransMIL [1] | 0.767 | 0.960 |
> | Ours | 0.895 | 0.971 |
>
> The two new datasets mentioned above are introduced in Section 8 in the Appendix of the revised manuscript, which is also shown below.
>
> The Cancer Genome Atlas colon and rectal cancer (TCGA-CRC-DX) [2] is used to test the performance of our SCL-WC method for microsatellite instability prediction, which contains a total of 428 WSIs (62 WSIs with microsatellite instability and 366 WSIs with microsatellite stability). The data splitting process is kept consistent with the original reference [2], i.e., 4-fold cross-validation based on the publicly released splitting table.
>
> TCGA-NSCLC is a subset of the TCGA dataset, which has been used for lung cancer subtyping: lung squamous cell carcinoma (TCGA-LUSC) and lung adenocarcinoma (TCGA-LUAD). It contains a total of 1043 diagnostic WSIs, including 531 WSIs with LUAD and 512 WSIs with LUSC. These WSIs are randomly divided into training, validation, and test sets at the patient level in the ratio of 65:10:25 [1].
>
> **While these datasets are relatively small.**
>
> We apologize for the confusion in our original writing, but we feel there may be some misunderstanding.
>
> (1) Compared to other state-of-the-art WSI weakly supervised classification methods in the literature, the WSI datasets used in our work are in fact much larger, containing a total of 15676 WSIs, which are in fact much larger. For example, the CLAM method uses five datasets containing a total of 3175 WSIs [4]. The TransMIL method uses three datasets with a total of 2267 WSIs [1]. Even in TCGA, the number of WSIs per cancer site also only varies from 200 to 2000, considering both frozen and diagnostic WSIs [16].
>
> (2) It is well known that tumors are highly variable and can have different appearances even for the same type of tumor [17]. Although we test our algorithm on WSI data from only two cancer sites, the variability across these datasets is very high. The two breast datasets are actually very different: Camelyon16 is for metastatic detection in breast cancer and BRACS is for 3-class breast tumor subtyping. The four prostate datasets aim to classify each image into two classes, but we adopt three of them as external test sets to validate the model generalizability to unseen data.

---

> > ### Author Response · Authors · 2022-08-02
> > **Response to Reviewer s5WU (2/5)**
> >
> > **The intuition behind the self-supervised feature extractor**
> >
> > We apologize for the lack of clarity in our reasoning to choose the Swin Transformer as the backbone model and MoCo v3 as the self-supervised learning framework. We have added these details in Section 3.1 in the Appendix in the revised manuscript, which are also shown below.
> >
> > *[Reasoning for using Swin Transformer]*: We choose the Swin Transformer as the backbone model due to the following considerations: (1) the Swin Transformer is considered as a generally applicable and the state-of-the-art backbone model for many computer vision problems. It contains a hierarchical ViT structure with local self-attention at each layer, which enables it to effectively extract multi-scale features both locally and from a global context [6]. It has been widely applied and shown to outperform CNN models in a variety of applications in both computer vision and medical imaging fields, such as image classification, semantic segmentation, and object detection [6][7]. (2) For histopathological image feature learning, it is also important to capture both local (cell-level structures) and global (tissue-level contexts) information [8]. Thus, the Swin Transformer model is a very suitable choice.
> >
> > *[Reasoning for using MoCo v3]*: We choose MoCo v3 as the self-supervised framework for two main reasons. First, MoCo v3 has been proven to outperform other self-supervised learning methods, such as the famous CNN-based SSL frameworks (e.g., SimCLR) [9] and the Transformer-based SSL frameworks (e.g., DINO) [10][11]. Second, it helps improve the stability of Transformer-type model pretraining and guarantees better performance.
> >
> > We also conducted some comparison experiments to verify the advantages of the MoCo v3 method, but we did not include them in the original manuscript due to page length limitations. The detailed results of these experiments can be seen in Table 2 in the Appendix of the revised manuscript, and also shown below.
> >
> > Table 2. Comparison of different self-supervised learning frameworks. Note that all methods use the same Swin Transformer as the backbone model.
> > | Methods | Camelyon16 |  | BRACS |  |
> > | :--- | :---: | :---: | :---: | :---: |
> > |  | ACC | AUC | ACC | AUC |
> > | SimCLR [12] | 0.8837 | 0.9344 | 0.7625 | 0.8488 |
> > | DINO [13] | 0.8992 | 0.9464 | 0.7875 | 0.8523 |
> > | MoCo v3 (Ours) | 0.9147 | 0.9566 | 0.8208 | 0.8650 |

---

> > > ### Author Response · Authors · 2022-08-02
> > > **Response to Reviewer s5WU (3/5)**
> > >
> > > **Why the authors have not provided how well the pre-trained feature extractor has worked in comparison to existing pre-trained models.**
> > >
> > > We have conducted new experiments to compare the method performance using different settings of feature extractors and feature aggregators, which are added in Section 9 in the Appendix of the revised manuscript and also shown below.
> > >
> > > (1) The proposed feature extractor is compared with existing feature extractors that are widely used for histopathological image analysis tasks, including an ImageNet-pretrained feature extractor and two other SSL-based feature extractors pretrained on histopathological images (SimCLR and DINO) [14][15]. To this end, we have conducted new experiments to investigate the performance of our SCL-WC method when using different feature extractors for microsatellite instability prediction on the TCGA-CRC-DX data and breast tumor subtyping on BRACS data. These existing feature extractors have been pretrained by previous studies and can be used as off-the-shelf feature extractors for comparison. Detailed results can be seen in Table 6 in the Appendix and are also shown below. It can be seen that the best performance is achieved by combining the proposed feature extractor with the proposed feature aggregator. Comparing different feature extractors tested on TCGA-CRC-DX data, it is also shown that our feature extraction method outperforms the previous best-performing DINO-ViT by around 7\% using both aggregation methods.
> > >
> > > (2) Table 6 also shows the effectiveness of the proposed feature aggregation scheme by keeping the same feature extractor while testing different feature aggregators (CLAM and ours). Comparing the corresponding columns of TCGA-CRC-DX data in Table 6, it is seen that our aggregation scheme outperforms that of the CLAM method by around 4\% for any of the four different feature extractors.
> > >
> > > Table 6: Results of weakly supervised classification using different feature extractors and feature aggregators. (Datasets: TCGA-CRC-DX and BRACS; Metric: AUC)
> > > | Feature extractors |  |  Feature aggregators |  |  |
> > > | :--- | :---: | :---: | :---: | :---: |
> > > |  | CLAM | Ours | CLAM | Ours |
> > > |  | TCGA-CRC-DX | TCGA-CRC-DX  | BRACS | BRACS |
> > > | ImageNet-ResNet50 | 0.7568 | 0.7806 | 0.8158 | 0.8446 |
> > > | SimCLR-ResNet18 [14] | 0.7716 | 0.8108 | 0.7634 | 0.8008 |
> > > | DINO-ViT [15] | 0.7853 | 0.8246 | 0.8108 | 0.8345 |
> > > | Ours-SwinTransformer | 0.8530 | 0.8954 | 0.8378 | 0.8650 |
> > >
> > >
> > > **Table 6 could have been extended ImageNet (P+N+HN) analysis as well.**
> > >
> > > The “ImageNet (P+N+HN)” refers to using our SCL-WC method but replacing the features with ImageNet-pretrained features. This experiment has been covered in Table 6 in the Appendix (i.e., the results in the fourth row). It is seen that our feature aggregation method combined with the ImageNet-pretrained features produces an AUC of 0.7806 on the TCGA-CRC-DX dataset, which is around 10\% lower than the full SCL-WC method shown in the last row of Table 6 in the Appendix.

---

> > > > ### Author Response · Authors · 2022-08-02
> > > > **Response to Reviewer s5WU (4/5)**
> > > >
> > > > **Difference between our attention pooling and state-of-the-art methodologies.**
> > > >
> > > > We apologize for the confusion in our original writing. In response, we will first introduce these state-of-the-art attention pooling methods, and then we will describe how our method differs from them. The detailed changes are highlighted in blue in Section 4 in the Appendix and are also shown below.
> > > >
> > > > (1) Among state-of-the-art weakly supervised WSI classification methods, attention pooling is an aggregation scheme widely used in the methods of ABMIL [3], DSMIL [5], and CLAM [4]. The ABMIL method is the first proposer of the attention pooling aggregator, which learns a class-agnostic weight for each patch and then performs a weighted element-wise summation to derive the final features. DSMIL is an extended version of the ABMIL method, which is improved by using different patch weights. It first uses a linear classifier to find a key patch with the highest score and then uses a distance measure between every patch and the key patch to obtain the weight for each patch. CLAM extends the ABMIL method to general multi-class weakly supervised classification tasks using a parallel attention structure for each class. It also adds an SVM-based instance classification branch to improve the instance discrimination ability. Even though these methods achieve superior performance, they never consider the global feature comparison across the training WSIs and the separation between positive and negative patches that are largely unbalanced within positive WSIs.
> > > >
> > > > (2) In our method, although the class-specific deep attention (CDA) module is similar to the CLAM method but with a different instance classification loss, our main contribution is the introduction of the positive-negative-aware modeling (PNM) and weakly-supervised cross-slide contrastive learning (WSCL) modules. The two novel modules fully consider the unique characteristics of WSIs to address the two limitations mentioned above. PNM uses the class-agnostic weight scores to separate the normal and abnormal (positive) regions within positive WSIs as much as possible. WSCL has no attention mechanism, but it helps to explore the complementary information within and across WSIs to obtain more discriminative feature representations, further improving the accuracy of the attention scores.
> > > >
> > > > **Implementation specifications.**
> > > >
> > > > We apologize for the overlook in our original writing. We have added the training details of our feature extractor and weakly supervised classification, including patch size and magnification information. The detailed changes are highlighted in blue in Table 1 in the revised manuscript and Section 3.2 and Section 5 in the Appendix, which are also shown below.
> > > >
> > > > (1) During pretraining, all WSIs in TCGA and PAIP are used as the training data, including frozen and diagnostic (formalin-fixed paraffin-embedded) slides. TCGA contains over 30,000 WSIs covering over 25 anatomical sites and 32 cancer subtypes [18]. PAIP contains 2457 WSIs covering six cancer types [19]. These WSIs are cropped into patches with a size of $512 \times 512$ pixels at 10$\times$ magnification, generating approximately 15 million patches. Our data augmentation strategies consider the characteristics of the histopathological images, including random cropping, Gaussian blur, and hue and saturation shifting in the HSV color space [20].
> > > >
> > > > (2) During weakly-supervised classification, these WSIs are cropped into nonoverlapping patches with a size of 224 $\times$ 224 pixels and a resolution of 1.0 microns per pixel (at around 10$\times$ magnification).

---

> > > > > ### Author Response · Authors · 2022-08-02
> > > > > **Response to Reviewer s5WU (5/5)**
> > > > >
> > > > >
> > > > > **References:**
> > > > >
> > > > > [1] Z. Shao, H. Bian, Y. Chen, Y. Wang, J. Zhang, X. Ji et al., “TransMIL: Transformer based correlated multiple instance learning for whole slide image classification,” NeurIPS, vol. 34, 2021.
> > > > >
> > > > > [2] M. Bilal, S. E. A. Raza, A. Azam, S. Graham, M. Ilyas, I. A. Cree, D. Snead, F. Minhas, and N. M. Rajpoot, “Development and validation of a weakly supervised deep learning framework to predict the status of molecular pathways and key mutations in colorectal cancer from routine histology images: a retrospective study,” The Lancet Digital Health, vol. 3, no. 12, pp. e763–e772, 2021.
> > > > >
> > > > > [3] M. Ilse, J. Tomczak, and M. Welling, “Attention-based deep multiple instance learning,” in ICML, 2018, pp. 2127–2136.
> > > > >
> > > > > [4] M. Y. Lu, D. F. Williamson, T. Y. Chen, R. J. Chen, M. Barbieri, and F. Mahmood, “Data efficient and weakly supervised computational pathology on whole-slide images,” Nature Biomedical Engineering, vol. 5, no. 6, pp. 555–570, 2021.
> > > > >
> > > > > [5] B. Li, Y. Li, and K. W. Eliceiri, “Dual-stream multiple instance learning network for whole slide image classification with self-supervised contrastive learning,” in CVPR, 2021, pp. 14318–14328.
> > > > >
> > > > > [6] Z. Liu, Y. Lin, Y. Cao, H. Hu, Y. Wei, Z. Zhang, S. Lin, and B. Guo, “Swin Transformer: Hierarchical vision transformer using shifted windows,” in ICCV, 2021, pp. 10012–10022.
> > > > >
> > > > > [7] F. Shamshad, S. Khan, S.W. Zamir, M. H. Khan, M. Hayat, F. S. Khan, and H. Fu, “Transformers in medical imaging: A survey,” arXiv preprint arXiv:2201.09873, 2022.
> > > > >
> > > > > [8] R. J. Chen, M. Y. Lu, W.-H. Weng, T. Y. Chen, D. F. Williamson, T. Manz, M. Shady, and F. Mahmood, “Multimodal co-attention transformer for survival prediction in gigapixel whole slide images,” in ICCV, 2021, pp. 4015–4025.
> > > > >
> > > > > [9] X. Chen, S. Xie, and K. He, “An empirical study of training self-supervised vision transformers,” in ICCV, 2021, pp. 9640–9649.
> > > > >
> > > > > [10] Z. Xie, Z. Zhang, Y. Cao, Y. Lin, J. Bao, Z. Yao, Q. Dai, and H. Hu, “SimMIM: A simple framework for masked image modeling,” in CVPR, 2022, pp. 9653–9663.
> > > > >
> > > > > [11] K. He, X. Chen, S. Xie, Y. Li, P. Dollár, and R. Girshick, “Masked autoencoders are scalable vision learners,” in CVPR, 2022, pp. 16 000–16 009.
> > > > >
> > > > > [12] T. Chen, S. Kornblith, M. Norouzi, and G. Hinton, “A simple framework for contrastive learning of visual representations,” in ICML, 2020, pp. 1597–1607.
> > > > >
> > > > > [13] M. Caron, H. Touvron, I. Misra, H. Jégou, J. Mairal, P. Bojanowski, and A. Joulin, “Emerging properties in self-supervised vision transformers,” in ICCV, 2021, pp. 9650–9660.
> > > > >
> > > > > [14] O. Ciga, T. Xu, and A. L. Martel, “Self supervised contrastive learning for digital histopathology,” Machine Learning with Applications, vol. 7, p. 100198, 2022.
> > > > >
> > > > > [15] R. J. Chen, C. Chen, Y. Li, T. Y. Chen, A. D. Trister, R. G. Krishnan, and F. Mahmood, “Scaling vision transformers to gigapixel images via hierarchical self-supervised learning,” in CVPR, 2022, pp. 16144–16155.
> > > > >
> > > > > [16] S. Kalra, H. R. Tizhoosh, S. Shah, C. Choi, S. Damaskinos, A. Safarpoor, S. Shafiei, M. Babaie, P. Diamandis, C. J. Campbell et al., “Pan-cancer diagnostic consensus through searching archival histopathology images using artificial intelligence,” NPJ Digital Medicine, vol. 3, no. 1, pp. 1–15, 2020.
> > > > >
> > > > > [17] R. Rashid, Y.A. Chen, J. Hoffer, J. L. Muhlich, J.R. Lin, R. Krueger, H. Pfister, R. Mitchell, S. Santagata, and P. K. Sorger, “Narrative online guides for the interpretation of digital-pathology images and tissue-atlas data,” Nature Biomedical Engineering, vol. 6, no. 5, pp. 515–526, 2022.
> > > > >
> > > > > [18] The Cancer Genome Atlas Project (TCGA). Available from: http://cancergenome.nih.gov/.
> > > > >
> > > > > [19] Y. J. Kim, H. Jang, K. Lee, S. Park, S. G. Min, C. Hong, J. H. Park, K. Lee, J. Kim, W. Hong et al., “PAIP 2019: Liver cancer segmentation challenge,” Medical Image Analysis, vol. 67, p. 101854, 2021.
> > > > >
> > > > > [20] D. Tellez, G. Litjens, P. Bándi, W. Bulten, J. M. Bokhorst, F. Ciompi, and J. Van Der Laak, “Quantifying the effects of data augmentation and stain color normalization in convolutional neural networks for computational pathology,” Medical Image Analysis, vol. 58, p. 101544, 2019.

---

> ### Author Response · Authors · 2022-08-09
> **Dear Reviewer s5WU,**
>
> We want to thank you again for your constructive comments to our original manuscript, and we have studied them thoroughly and made corresponding responses and revisions. With the end of the discussion period approaching, we would like to summarize our responses again. We would appreciate it very much if you could take a look to see whether we have sufficiently addressed your original concerns.
>
> (1) We have added explanations about the intuition behind the self-supervised feature extractor, including the backbone model and the self-supervised learning framework. Also, more comparison experiments are conducted to verify our reasoning.
>
> (2) We have added new experiments to compare the proposed feature extractor with existing feature extractors that are widely used for histopathological image analysis tasks.
>
> (3) We have added two new weakly supervised WSI classification experiments on two more cancer types (i.e., TCGA-CRC-DX for microsatellite instability prediction of colorectal cancer and TCGA-NSCLC for lung tumor subtyping).
>
> (4) We have clarified that the previous validation datasets are pretty large in size, even though only containing two cancer types.
>
> (5) We have added the training details of our feature extractor and weakly supervised classification method, including patch size and magnification information.
>
> (6) We have added an explanation about the differences between our attention pooling approach and the closely related state-of-the-art methodologies.
>
> Thank you again for your time and patience. We would greatly appreciate your second feedback to ensure that our responses and revisions have adequately addressed your original concerns. Please let us know if there are other issues that could be addressed.
>
> Sincerely,
>
> Authors of Paper4284

---

### Official Review · Reviewer_B23m · 2022-07-11

**Rating:** 7
**Confidence:** 3
**Soundness:** 3 good
**Presentation:** 3 good
**Contribution:** 3 good

**Summary:**

The paper proposed a novel framework for Weakly-Supervised Whole-Slide Image Classification. It uses a self-supervised contrastive representation encoder trained with large-scale data. It implements three components to aggregate the obtained patch-level representations:  deep attention-based class-specific MIL aggregation(CDA), a positive-negative-aware model that pushes away positive and negative patches given its relevance to the slide-level prediction (PNM), and a weakly-supervised cross-slide contrastive learning with both hard negative and negative sample mining.


**Questions:**

1. How is the class-agnostic weight score computed precisely in line 163?
2. How are the hyperparameters selected ($\alpha$, et.al)?
3. The presentation of tables on page 7 needs refinements.

**Limitations:**

The author mentioned about validating in larger cohorts from real-world clinical setting as future work.

**Strengths And Weaknesses:**

* The paper adapted existing ideas to a different application given the characteristics of the problem they aim to solve.
* The paper is well written and easy to follow. The main contributions in terms of the method are well-explained.
* The proposed methods are well justified and intuitively understandable. They also achieved very strong performance on multiple datasets.
* Ablation studies are conducted to help understand the importance of each proposed component.

---

> ### Author Response · Authors · 2022-08-02
> **Response to Reviewer B23m**
>
> We thank the reviewer for the careful review of our manuscript and the many constructive and positive comments. We are very pleased to see that the reviewer agrees our method is well-explained with clear presentation and strong performance. We apologize for the lack of clarity on some parts of our original manuscript, as pointed out by the reviewer. We have added more explanations in the one-by-one responses below and will also revise the manuscript accordingly. We hope these responses can help clarify the issues.
>
> **How is the class-agnostic weight score computed precisely in line 163?**
>
> We apologize for the lack of clarity. The description and the equation below explain the detailed computational process for the class-agnostic weight score, which will be also added to the revised manuscript (“Positive-negative-aware modeling” paragraph of Section 3).
>
> The class-agnostic weight score in Eq. (5) is computed through two fully connected layers: “Specifically, we first characterize the relevance of the $l^{th}$ patch to the slide-level prediction by calculating its class-agnostic weight score $\tilde{A}_{n}^{l}$
> through two fully connected layers.” (The detailed equation can be seen in the revised manuscript.)
>
> **How are the hyperparameters selected?**
>
> We apologize for the lack of clarity in hyperparameter selection in Section 5 of the Appendix in the original manuscript. In fact, $\beta$ and $\gamma$ in $\mathcal{L}_{\mathrm{total}}$ are determined by ablation experiments. We have added the experiment results in Table 3 in Section 5 of the Appendix, which are also shown below.
>
> Table 3. Effect of hyperparameters $\beta$ and $\gamma$ on classification accuracy of the Camelyon16
> dataset (AUC)
> | $\beta$ |  | $\gamma$ |  |
> | :--- | :--- | :--- | :--- |
> |  | 0.001 | 0.01 | 0.1 |
> | 0.02 | 0.9372 | 0.9464 | 0.9418 |
> | 0.2 | 0.9497 | 0.9566 | 0.9464 |
>
> It can be seen that our method is pretty stable for different choices of the two hyperparameters. Slightly better results are obtained for $\beta$=0.2 and $\gamma$=0.01, which are used for all experiments reported in the paper.
>
> **The presentation of tables on page 7 needs refinements.**
>
> We thank the reviewer for pointing out the problem. We have rearranged the presentation of these Tables in page 7.

---

### Official Review · Reviewer_oepB · 2022-07-12

**Rating:** 5
**Confidence:** 5
**Soundness:** 3 good
**Presentation:** 3 good
**Contribution:** 3 good

**Summary:**

This paper proposed a novel weakly-supervised whole-slide image classification method called SCL-WC, which is constructed based on a domain-specific self-supervised learning feature extractor and a task-specific feature aggregator. The proposed SCL-WC method outperforms state-of-the-art weakly-supervised whole-slide image classification studies over four publicly available datasets for the binary/multiple classification tasks.

**Questions:**

1. Why use swin transformer as the feature encoder?
2. Why use MoCo V3 as self supervised learning? what is the detailed implementation process of Moco V3 in the whole framework ?

**Limitations:**

see weakness

**Strengths And Weaknesses:**

Strength：
1. The overall framework is novel and interesting. The use of positive-negative-aware module (PNM) and a weakly-supervised cross-slide contrastive learning (WSCL) module to achieve both intra-WSI local patch separation and inter-WSI global feature contrast is smart.
2. The motivation and introduction are clear.
3. The multi-class classification used in multi-instance learning is novel.

Weakness：
3. The application of contrastive learning, i.e., MoCov3, in histopathological image classification is very common.
4. The assignment of the three hyper-parameter α，β，γ of the loss function mentioned in this paper lacks sufficient experiments.

---

> ### Author Response · Authors · 2022-08-02
> **Response to Reviewer oepB (1/2)**
>
> We thank the reviewer for the careful review and constructive feedback. We are very pleased to see that the reviewer finds our method novel, clear, and interesting. We apologize for the lack of clarity on some parts of our original manuscript, as pointed out by the reviewer. Below are our one-by-one responses to address these concerns of the reviewer, which we hope can clarify the issues.
>
> **Why use Swin Transformer as the feature encoder?**
>
> We would like to apologize for not giving enough reasoning for choosing the Swin Transformer as the backbone model. The following explanation will be added to the revised manuscript in Section 3.1 of the Appendix.
>
> We choose the Swin Transformer as the backbone model due to the following considerations: (1) the Swin Transformer is considered as a generally applicable and the state-of-the-art backbone model for many computer vision problems. It contains a hierarchical ViT structure with local self-attention at each layer, which enables it to effectively extract multi-scale features both locally and from a global context [A]. It has been widely applied and shown to outperform CNN models in a variety of applications in both computer vision and medical imaging fields, such as image classification, semantic segmentation, and object detection [A][B]. (2) For histopathological image feature learning, it is also important to capture both local (cell-level structures) and global (tissue-level contexts) information [C]. Thus, the Swin Transformer model is a very suitable choice.
>
> **Hyperparameters:**
> We apologize for the lack of clarity in hyperparameter selection in Section 5 of the Appendix in the original manuscript. In fact, $\beta$ and $\gamma$ in $\mathcal{L}_{\mathrm{total}}$ are determined by ablation experiments. We have added the experiment results in Table 3 in Section 5 of the Appendix, which are also shown below.
>
> Table 3. Effect of hyperparameters $\beta$ and $\gamma$ on classification accuracy of the Camelyon16
> dataset (AUC)
> | $\beta$ |  | $\gamma$ |  |
> | :--- | :--- | :--- | :--- |
> |  | 0.001 | 0.01 | 0.1 |
> | 0.02 | 0.9372 | 0.9464 | 0.9418 |
> | 0.2 | 0.9497 | 0.9566 | 0.9464 |
>
> It can be seen that our method is pretty stable for different choices of the two hyperparameters. Slightly better results are obtained for $\beta$=0.2 and $\gamma$=0.01, which are used for all experiments reported in the paper.
>
> References:
>
> [A].	Z. Liu, Y. Lin, Y. Cao, H. Hu, Y. Wei, Z. Zhang, S. Lin, and B. Guo, “Swin Transformer: Hierarchical vision transformer using shifted windows,” in ICCV, 2021, pp. 10012–10022.
>
> [B].	F. Shamshad, S. Khan, S.W. Zamir, M. H. Khan, M. Hayat, F. S. Khan, and H. Fu, “Transformers in medical imaging: A survey,” arXiv preprint arXiv:2201.09873, 2022.
>
> [C].	R. J. Chen, M. Y. Lu, W.-H. Weng, T. Y. Chen, D. F. Williamson, T. Manz, M. Shady, and F. Mahmood, “Multimodal co-attention transformer for survival prediction in gigapixel whole slide images,” in ICCV, 2021, pp. 4015–4025.

---

> > ### Author Response · Authors · 2022-08-02
> > **Response to Reviewer oepB (2/2)**
> >
> > **Why use MoCo v3 as self-supervised learning? What is the detailed implementation process of MoCo v3 in the whole framework?**
> >
> > We apologize for the lack of clarity in our reasoning to choose MoCo v3 as the self-supervised learning framework and in the implementation process of MoCo v3. We have added these details in Section 3.1 and Section 3.2 in the Appendix of the revised manuscript, which are also shown below.
> >
> > *[Reasoning for using MoCo v3]*: We choose MoCo v3 as the self-supervised framework for two main reasons. First, MoCo v3 has been proven to outperform other self-supervised learning methods, such as the famous CNN-based SSL frameworks (e.g., SimCLR) [D] and the Transformer-based SSL frameworks (e.g., DINO) [E][F]. Second, it helps improve the stability of Transformer-type model pretraining and guarantees better performance [D].
> >
> > We also conducted some comparison experiments to verify the advantages of the MoCo v3 method, but we did not include them in the original manuscript due to page length limitations. The detailed results of these experiments can be seen in Table 2 in the Appendix of the revised manuscript, and also shown below.
> >
> > Table 2. Comparison of different self-supervised learning frameworks. Note that all methods use the same Swin Transformer as the backbone model.
> > | Methods | Camelyon16 |  | BRACS |  |
> > | :--- | :---: | :---: | :---: | :---: |
> > |  | ACC | AUC | ACC | AUC |
> > | SimCLR [G] | 0.8837 | 0.9344 | 0.7625 | 0.8488 |
> > | DINO [H] | 0.8992 | 0.9464 | 0.7875 | 0.8523 |
> > | MoCo v3 (Ours) | 0.9147 | 0.9566 | 0.8208 | 0.8650 |
> >
> > *[Implementation details of MoCo v3]*: MoCo v3 uses two parallel branches (online and target branches) to learn meaningful feature representations by solving a pretext task of instance discrimination. These two branches have similar encoder models ($f_q$ and $f_k$) and MLP projectors ($h_q$ and $h_k$). The $f_k$ is momentum updated from $f_q$. The target branch has another MLP projector $g_k$ following $h_k$. Given an image, its two augmented views are used as inputs to the two branches, which are then encoded and projected by the corresponding feature encoders and MLP projectors to obtain the final feature vectors $z_q$ and $z_k$. A contrastive loss is used to pull features obtained from the same image (e.g., $z_q$ and $z_k$) together and push apart features obtained from different images. In our work, we use the Swin Transformer as the backbone model. The first MLP projector ($h_q$ and $h_k$) and the second MLP projector ($g_k$) have three and two linear layers, respectively.
> >
> > During pretraining, all WSIs in TCGA and PAIP are used as the training data, including frozen and diagnostic (formalin-fixed paraffin-embedded) slides. TCGA contains over 30,000 WSIs covering over 25 anatomical sites and 32 cancer subtypes [I]. PAIP contains 2457 WSIs covering six cancer types [J]. These WSIs are cropped into patches with a size of $512 \times 512$ pixels at 10$\times$ magnification, generating approximately 15 million patches. Our data augmentation strategies consider the characteristics of the histopathological images, including random cropping, Gaussian blur, and hue and saturation shifting in the HSV color space [K].
> >
> > References:
> >
> > [D].	X. Chen, S. Xie, and K. He, “An empirical study of training self-supervised vision transformers,” in ICCV, 2021, pp. 9640–9649.
> >
> > [E].	Z. Xie, Z. Zhang, Y. Cao, Y. Lin, J. Bao, Z. Yao, Q. Dai, and H. Hu, “SimMIM: A simple framework for masked image modeling,” in CVPR, 2022, pp. 9653–9663.
> >
> > [F].	K. He, X. Chen, S. Xie, Y. Li, P. Dollár, and R. Girshick, “Masked autoencoders are scalable vision learners,” in CVPR, 2022, pp. 16 000–16 009.
> >
> > [G].	T. Chen, S. Kornblith, M. Norouzi, and G. Hinton, “A simple framework for contrastive learning of visual representations,” in ICML, 2020, pp. 1597–1607.
> >
> > [H].	M. Caron, H. Touvron, I. Misra, H. Jégou, J. Mairal, P. Bojanowski, and A. Joulin, “Emerging properties in self-supervised vision transformers,” in ICCV, 2021, pp. 9650–9660.
> >
> > [I].	The Cancer Genome Atlas Project (TCGA). Available from: http://cancergenome.nih.gov/.
> >
> > [J].	Y. J. Kim, H. Jang, K. Lee, S. Park, S. G. Min, C. Hong, J. H. Park, K. Lee, J. Kim, W. Hong et al., “PAIP 2019: Liver cancer segmentation challenge,” Medical Image Analysis, vol. 67, p. 101854, 2021.
> >
> > [K].	D. Tellez, G. Litjens, P. Bándi, W. Bulten, J. M. Bokhorst, F. Ciompi, and J. Van Der Laak, “Quantifying the effects of data augmentation and stain color normalization in convolutional neural networks for computational pathology,” Medical Image Analysis, vol. 58, p. 101544, 2019.

---

### Author Response · Authors · 2022-08-08
**A gentle reminder for the end of the discussion period**

Dear Reviewers,

Thanks again for your valuable comments to improve our manuscript. We are leaving a gentle reminder for the end of the discussion period. We would like to know if we have addressed all concerns raised by the reviewers in our first feedback, or if there are additional issues that could be addressed.

Sincerely,

Authors of Paper4284

---

### Meta-Review · Area_Chair_s4PP · 2022-08-26

**Recommendation:** Accept
**Confidence:** Less certain

**Metareview:**

This work presents a method to obtain slide level representations in computational pathology. The specific contributions of this work are a positive-negative-aware module (PNM) and a weakly-supervised cross-slide contrastive learning (WSCL) module and a loss to encourage intra-WSI local patch separation and inter-WSI global feature contrast. The idea is to use the attention weights in a MIL framework as patch level pseudo labels. These are used to compute "weights" for positive and negative patches (the latter of which is assumed to be significantly larger in number for a given WSI). By using these weights in a contrastive manner to push the representations from positive patches away from those of negative patches, the model learns more effectively since it is less susceptible to the noise from the negative patches.

The reviewers found these contributions novel. During the review, the largest source of concern was along the choices made during the empirical evaluation. Specifically, the manuscript in its current form lacked empirical backing for several of the choices made regarding the neural architecture, the algorithm for self-supervised learning etc. In response to this the authors conducted several different kinds of ablation studies (which were incorporated into the supplement) during the rebuttal process, which other reviewers found convincing as a potential explanation of the outcomes.

Overall, I found the main contributions of this work (leveraging patch level attention weights as psuedo labels) to be an interesting use case for computational pathology to better focus the learning signal on positive patches. My additional comment is that I think the additional ablation experiments (and references) are an important part of the contributions of this work and should be incorporated into the main paper rather than in the appendix (several equations can be compressed to make space in the manuscript in addition to the additional page).

**Award:**

No

---

### Decision · Program_Chairs · 2022-09-14

Accept